# Prediction Equations of the Multifrequency Standing and Supine Bioimpedance for Appendicular Skeletal Muscle Mass in Korean Older People

**DOI:** 10.3390/ijerph17165847

**Published:** 2020-08-12

**Authors:** Kwon Chan Jeon, So-Young Kim, Fang Lin Jiang, Sochung Chung, Jatin P. Ambegaonkar, Jae-Hyeon Park, Young-Joo Kim, Chul-Hyun Kim

**Affiliations:** 1Department of Health and Human Performance, Northwestern State University, Natchitoches, LA 71497, USA; kcjeon@nsula.edu; 2Department of Food Science & Nutrition, Soonchunhyang University, Asan 31438, Korea; sonyah@sch.ac.kr; 3Department of Sports Medicine, Soonchunhyang University, Asan 31438, Korea; jfl0108681@gmail.com; 4Department of Pediatrics, Konkuk University Medical Center, School of Medicine, Konkuk University, Seoul 413-729, Korea; scchung@kuh.ac.kr; 5SMART Laboratory, School of Kinesiology, George Mason University, Manassas, VA 10890, USA; jambegao@gmu.edu; 6Department of Sport and Health Aging, Korea National Sport University, Seoul 05541, Korea; jhpark@knsu.ac.kr; 7Department of Exercise Rehabilitation Welfare, Sungshin Women’s University, Seoul 02844, Korea; kyj87@sungshin.ac.kr

**Keywords:** multifrequency bioelectrical impedance analysis (MF-BIA), dual-energy X-ray absorptiometry (DXA), clinical diagnosis, sarcopenia, appendicular skeletal muscle mass

## Abstract

Bioimpedance analysis (BIA) has been demanded for the assessment of appendicular skeletal muscle mass (ASM) in clinical and epidemiological settings. This study aimed to validate BIA equations for predicting ASM in the standing and supine positions; externally to cross-validate the new and published and built-in BIA equations for group and individual predictive accuracy; and to assess the overall agreement between the measured and predicted ASM index as sarcopenia diagnosis. In total, 199 healthy older adults completed the measurements of multifrequency BIA (InBody770 and InBodyS10) and dual-energy X-ray absorptiometry (DXA). Multiple regression analysis was used to validate the new multifrequency bioelectrical impedance analysis (MF-BIA) prediction equations. Each MF-BIA equation in the standing and supine position developed in the entire group included height^2^/resistance, sex, and reactance as predictors (*R*^2^ = 92.7% and 92.8%, SEE = 1.02 kg and 1.01 kg ASM for the standing and supine MF-BIA). The new MF-BIA equations had a specificity positive predictive value and negative predictive value of 85% or more except for a sensitivity of about 60.0%. The new standing and supine MF-BIA prediction equation are useful for epidemiological and field settings as well as a clinical diagnosis of sarcopenia. Future research is needed to improve the sensitivity of diagnosis of sarcopenia using MF-BIA.

## 1. Introduction

Sarcopenia is a condition that involves a loss of skeletal muscle mass in aging. Due to this drastic reduction in muscle mass, physical functional impairment and disability occur [1]. The World Health Organization (WHO) assigned the disease code ICD-10-CM (M62.84) to sarcopenia [2]. Unfortunately, sarcopenia research is at least 15–20 years behind osteopenia or osteoporosis research [3]. Based on the findings from the New Mexico Elderly Health Survey in 1998, Baumgartner et al. first proposed diagnostic criteria for sarcopenia using appendicular skeletal muscle mass (ASM) [4]. As a result, diagnostic criteria published by organizations such as the European Working Group on Sarcopenia in Older People (EWGSOP), International Working Group on Sarcopenia (IWGS) and Asian Working Group on Sarcopenia (AWGS) included the concept of muscle function, which refers to muscle strength and/or performance ability, starting from 2010 [5,6,7,8,9].

The most accurate method for measuring muscle mass available today is computed tomography (CT) or magnetic resonance imaging (MRI). However, the use of CT or MRI is recommended only for research purposes, because CT produces high dose radiation exposure and requires high cost and a fairly large fixed space and MRI is highly expensive and occupies a large space. In clinical practice, the dual energy X-ray absorptiometry (DXA) method has received much attention, as its results highly correlate with MRI measurements while exposing patients to much lower doses of radiation while offering the convenience of testing. DXA measures the body composition of bone mineral content, body fat mass, and lean body mass from each part of the body. The sum of muscle mass calculated by subtracting bone mineral content from lean body mass of both arms and legs (e.g., the appendages) is referred to as appendicular skeletal muscle mass (ASM), which is used as a major indicator of sarcopenia [10,11].

Despite the numerous advantages of using DXA, it is not suitable in all situations to measure body composition, because a DXA is a large and expensive machine placed in a fixed location. For these reasons, use of DXA in a large number of epidemiological studies is limited. To overcome these issue, bioelectric impedance analysis (BIA) is considered as an alternative option in large-scale population studies. BIA estimates muscle mass based on the magnitude of impedance from a micro-current flowing through the body. In BIA, a low electrical current is sent through the body to estimate body water and cell membrane proportions. Based on this, the volume of total body water and body cell mass can then be measured. Subsequently, muscle mass can be estimated, as most of the body’s water is found in its muscles. Due to practical benefits of using BIA (e.g., non-invasive, low cost and ease of use), it is a valuable piece of equipment for epidemiological and clinical settings [10,11,12].

In recent years, several studies have used BIA measurements to estimate ASM and develop BIA prediction equation [6,8,12,13,14,15,16,17,18,19,20,21]. Most of these equations have been developed from a single frequency BIA (SF-BIA) at 50 kHz to assess ASM in a supine position. However, SF-BIA (i.e., 50 kHz) has not been shown to detect changes in body composition when these changes coincide with alternations in the body water distribution [11]. This effect can be ascribed to a lower specific resistivity of the intracellular fluids, such as from water-loss dehydration in older people. The dehydration causes a cascade effect of increasing the specific resistivity of the extracellular fluid (ECF), resulting in a high than expected increase in body impedance, and thus an inability to detect intracellular fluid in tissues such as appendicular skeletal muscles [11,12]. Conversely, multifrequency BIA (MF-BIA) devices can overcome the problems encountered with single-frequency BIA by using low and high-frequency electric current in the range of 1~1000 kHz as the low frequency has high conductivity in extracellular fluid and the high frequency has high conductivity in intracellular fluid [11,12,13].

Studies that developed the prediction equations for ASM with MF-BIA devices have been reported by Kim, et al. (2014) [18] and Yamada, et al. (2017) [20]. These studies developed and cross-validated prediction equations of ASM using MF-BIA with a large population of older Koreans or older Japanese. However, the two studies predicted the descriptive statistics of the ASM in the Korean elderly group and the descriptive statistics of the ASM in the Japanese elderly group, but not the individual ASM of the elderly. Sarcopenia societies recommend the use of BIA not only for large-scale epidemiological studies but also for clinical applications requiring individual assessment [5,6,7,8]. Therefore, it is necessary to develop MF-BIA prediction equations that allow MF-BIA to examine ASM at individual levels as well as in groups. The MF-BIA devices reported so far have been developed as a standing type, and recently, a new supine type of the MF-BIA was introduced. The human body can change the distribution of body water depending on the posture, and as a result, the impedance when standing and the impedance when lying must be different [12,22], so the prediction equations must be developed separately.

Therefore, the aims of this study were (1) to develop and cross-validate MF-BIA equations for predicting ASM in the standing and supine positions, (2) externally to cross-validate new and published BIA-based (and built-in) equations for the group and individual predictive accuracy, and (3) to assess the predictive agreement between BIA- and DXA-derived appendicular skeletal muscle index (ASMI) as sarcopenia diagnosis in terms of the Asian Working Group on Sarcopenia (AWGS) cut-offs [9] using DXA as a reference method in the community-dwelling older Korean population.

## 2. Materials and Methods

### 2.1. Subjects

One hundred ninety-nine community dwelling older Korean participants (94 men and 105 women) aged 70–92 years were included in this study. Subjects were recruited through advertisements in local newspapers, social media, and invitations to participate in the study were sent to elderly members of public welfare communities. Exclusion criteria were critical or terminal illness, skeletal muscle diseases, hospitalization within 3 months, more than 6 kg weight loss within 6 months of measurement, electrical or metallic implant (expect tooth implants) and complete or partial amputation of one or more limb. All participants gave written informed consent prior to participating in the study. The study was approved by the ethical committee of Korean National Sport University (No.1263-201903-HR-010-02) and performed in accordance with the Declaration of Helsinki. Subjects refrained from alcohol intake for 12 h, fasted at least 4 h and voided before undergoing measurements at the Body Composition Laboratory. Body weight was measured using a scale (model DB-1, CAS, South Korea) to the closest of 0.05 kg with subjects in light clothing. Heights were measured to nearest 0.1 cm with a seca 274 stadiometer (seca GmbH, Hamburg, Germany).

### 2.2. Appendicular Skeletal Muscle Mass from Dual-Energy X-Ray Absorptiometry

A whole-body DXA Prodigy Advance scanner (GE Lunar, Madison, WI, USA) was used to measure each participant’s total and appendicular bone mineral content, fat mass, and lean mass. The DXA instrument was calibrated daily using the spine phantom provided by the manufacture. For standardization purposes of the scans, the files from the original DXA machine were transferred to iDXA Software version 4.0.2. Whole body scans were manually analyzed for the manufacture defined regions of interest (ROI) following the standard analysis protocol in the GE Lunar User Manual. Customized ROI also analyzed according to Hyemsfield’s protocol [23] using the whole body and subregion analysis mode. According to the Hyemsfield’s protocol, the boundaries of the regions of interest (ROIs) were defined as follows: (1) for the upper limbs of the ROIs (right and left), the arms are isolated by running a line through the humeral head and (2) for the lower limbs, the pelvis cut is placed just above the pelvic brim and the system computer automatically draws the lower pelvic lines to bisect the hip joints.

### 2.3. Multifrequency 8-Electrodes Bioimpedance Analysis

Two multifrequency 8-electrode BIAs (InBody 770 for the standing measure and the InBody S10 for the supine measure, InBody Co. Seoul, South Korea) were performed for this study. This BIA model uses eight electrodes positioned at each hand and foot and enable multifrequency impedance measurement of the arms, trunk and legs. Impedance parameters were measured with an alternating current of 80 mA and 100 mA at the frequency of 5–1000 kHz for InBody S10 and InBody770.

For the standing measure of InBody770, the participants were instructed to stand with her or his soles in contact with the foot electrodes and grab the hand electrodes as shown in Figure 1A. InBody S10 is designed for measurements in the supine position and operated using eight electrodes on right and left sides while lying on a non-conductive surface. Adhesive gel electrodes were placed at defined anatomical sites on the dorsal surfaces of the hand, wrist, ankle and foot as follows: the proximal edge of the wrist electrode was attached from an imaginary line bisecting the styloid process of the ulna and the proximal edge of the finger electrode from an imaginary line bisecting the metatarsophalangeal joint of the middle finger. The proximal edge of the ankle electrode was attached from an imaginary line bisecting the medial malleolus and the distal edge of the toe electrode was placed from an imaginary line through the metatarsophalangeal joints of the second toe, as shown in Figure 1B. Resistance [*R*] and reactance [*Xc*] were measured with the two multiple-frequency BIAs, InBody770 and InBody S10 at six different frequencies (1, 5, 50, 250, 500, and 1000 kHz). Impedance index was calculated as height^2^/resistance. The devices were calibrated every morning using the standard control circuit supplied by the manufacture. The precision error of fat-free mass is less than 2%.

### 2.4. Published Prediction Equations

The 50 kHz-frequency BIA-based equations were used for the external cross-validation and the agreement for the classification of sarcopenia in our participants. ASM was calculated using two built-in equation from InBody770 and InBody S10 and the following eight equations including Vermeiren et al. (2019) [14], Scafoglieri et al. (2017) [15], Sergi et al. (2015) [16], Kyle et al. (2003) [17], Kim et al. (2014) [18], Peniche et al. (2015) [19], Yamada et al. (2017) [20], respectively:

ASM_Vermeiren_ (kg) = 0.827 + 0.19Ht^2^*/Z* + 2.101Sex + 0.079Wt; (*R*^2^ = 0.888, SEE = 1.45 kg)


ASM_Scafoglieri_ (kg) = 1.821 + 0.168 Ht^2^/*R* + 0.132Wt − 1.931Sex + 0.017Xc; (*R*^2^ = 0.89, SEE = 1.45 kg)


ASM_Sergi_ (kg) = 3.964 + 0.227Ht^2^/*R* + 0.095Wt + 1.384Sex + 0.064Xc; (*R*^2^ = 0.92, SEE = 1.14 kg)


ASM_Kyle_ (kg) = 4.211 + 0.267Ht^2^/*R* + 0.095Wt + 1.909Sex − 0.012age; (*R*^2^ = 0.95, SEE = 1.12 kg)


ASM_Kim_ (kg) = 5.663 + 0.104Ht^2^/*R* − 0.050age + 2.954Sex + 0.055Wt; (*R*^2^ = 0.88, SEE = 1.35 kg)


ASM_Peniche_ (kg) = −0.05376 + 0.2394Ht^2^/*R* + 2.708Sex + 0.065Wt; (*R*^2^ = 0.91, SEE = 1.01 kg)


ASM_Yamada_for_men_ (kg) = 51.33 + 0.6947*ZI*_@50kHz_) − 55.24(*Z*_@z250kHz_/*Z*_@5kHz_) − 10940(1/Z_@50kHz_); (*R*^2^ = 0.87, SEE = 1.46 kg)


ASM_Yamada_for_women_ (kg) = 37.91 + 0.6144*ZI*_@50kHz_ − 36.61(*Z*_@z250kHz_/*Z*_@5kHz_) − 9332(1/Z_@50kHz_); (*R*^2^ = 0.86, SEE = 1.22 kg)

where Ht is height in centimeters; Wt is body weight in kg; age is in years; *R* is resistance, *Xc* is reactance, *ZI*_@50kHz_ is the impedance index at 50 kHz, *Z*_@5kHz_ is impedance at 5 kHz, *Z*_@50kHz_ is impedance at 50 kHz, *Z*_@250kHz_ is impedance at 250 kHz derived from BIA; for sex men = 1 and women = 0.

### 2.5. Statistical Analysis

The physical characteristics of the development and cross-validation groups are presented as means with SDs. Independent *t*-tests were used to determine significant differences between mean values within each sex between groups (development and cross-validation group). The stepwise multiple linear regression analysis was used to develop the ASM-estimating equations in the development group. Variables included in the initial analyses contained *RI*_@50kHz_, *RI*_@50kHz_, *RI*_@250kHz_, *Xc*_@5kHz_, *Xc*_@50kHz_, age (yr), weight (kg) and sex (dummy coded with women = 0 and men = 1). These equations were developed on 2/3 of the total sample selected randomly (development sample) with the remaining 1/3(validation sample) used for cross-validation. The developmental equations were selected by measures of goodness-of-fit statistics, including *r*^2^, the standard error of estimate (SEE), acceptable subjective rating of SEE (i.e., good to excellent) according to the minimally acceptable standard for prediction errors [11,24,25], the coefficient of variation (CV), and the variance inflation factor (VIF). The SEE measures the variation in the actual values from the predicted values. The SEE represents the degree of deviation of individual scores form the regression line. It is computed as the following formula: SEE = $\sqrt{\sum {\left(Measured\;ASM-Estimated\;ASM\right)}^{2}/\left(N-p-1\right)}$ where *p* = number of predicter variables. The CV is a measure of dispersion of the SEE from the mean of the actual values for the accuracy that is computed as following: CV = (SEE ÷ mean of the DXA-measured ASM) × 100. The VIF assesses how much the variance of an estimated regression coefficient increases when predictors are correlated for estimating collinearity/multicollinearity. Higher values of more than 10 can be assumed that the regression coefficients are poorly estimated due to muticollinearity to remove predictors from the model. In our study, with values less than 10, we are in good shape and can process with our regression. In the internal cross-validation, the group predictive accuracy of the ASM_BIA_ equations was tested by calculating *r*^2^, total error (TE: The TE represents the degree of deviation from the line of identity using the formula: Total Error = $\sqrt{\sum {\left(MeasuredaASM-EstimatedaASM\right)}^{2}/N}$), and acceptable subjective rating of TE. The individual predictive accuracy of these equations was also tested by Bland–Altman plots that includes the constant error (CE; the bias of the mean difference between measured and predicted values) tested against zero using paired *T*-Test, 95% limits of agreement between equations, concordance correlation efficient (*r*_y-y’,mean_) and percentage of individual agreement (PIA). In the PIA, the minimum acceptable standard for prediction errors within ±1.45 kg and ±1.16 kg SEE or TE of ASM for men and women (i.e., good rating) [11,24,25] is plotted on the Bland–Altman plot and the percentage of individuals falling inside of these limits is calculated as following: PIA = (the number of residual scores within ±1.45 kg for men and ±1.16 kg for women ÷ total residual scores) × 100. The residual score is an individual difference between a DXA-measured ASM and a BIA-predicted ASM. For the external cross-validation study, the group and individual predictive accuracy were calculated for the published and built-in BIA equations. Overall agreement for the classification of sarcopenia by BIA equations and DXA measurements was performed by the 2 × 2 contingency table using a Cohen’s Kappa. The 5% level was chosen for statistical significance. Data were analyzed using SPSS version 21.0 (IBM, USA).

## 3. Results

### 3.1. Characteristics of the Study Population

One hundred and ninety-nine (94 men, 105 women) older adults participated in the present study. The total sample was split randomly into groups of 133 (development sample) and 66 (internal cross-validation sample), to develop BIA prediction equations for ASM. General characteristics for the two study groups are shown in Table 1. The between-groups had no difference in all the variables and the between-sex had no difference in Age, BMI, and *Xc*. Men were taller, weighed more, had more fat-free mass (FFM), ASM, higher *ZI* and *RI* and less fat mass (FM), percentage of body fat (PBF), *Z*, and *R* than women in the development and cross-validation group (all *p* < 0.05).

### 3.2. Development and Cross-Validation of BIA Prediction Equations for ASM

The total sample was randomly assigned into the development group (*n* = 133, 67%) and the cross-validation group (*n* = 66, 33%) to develop BIA prediction equations for ASM based on BIA measurements in the standing or supine position. For the development group, the relationship between ASM as the dependent variable and *RI*_@50kHz_, *RI*_@250kHz_, *RI*_@250kHz_, *Xc*_@50kHz_, *Xc*_@5kHz_, age, sex, and body weight as the dependent variables were analyzed through the stepwise multiple regression model. Within the standing and supine BIA measurements, *RI*, *Xc*, and sex (with body weight) were significant contributors to the BIA best-fitting regression model with maximum adjusted *R*^2^ = 92.1% and 91.7% and minimum SEE = 1.06 kg and 1.08 kg for the standing vs. supine BIA regression equation, which resulted in the same subjective rating (men = “excellent”, women = “good”) as shown in Table 2. The coefficients of variation (CV) for the two regression equations were 6.4% and 6.1%, respectively. The variance inflation factor (VIF) of all independent variables in each regression equation was less than 5 with no multicollinearity among variables. The two regression equations for the development group were used to predict ASM in the cross-validation group. In Table 2, the results showed that there was no significant mean difference in ASM between the DXA measure and each BIA prediction in the standing and supine equation. The group predictive accuracy of *R*^2^, TE and subjective rating was given as 93.4% and 1.00 kg with the “excellent” in men and “very good” rating in women and 94.8% and 0.93 kg with the “excellent” in men and “very good” rating in women for the standing and supine BIA prediction equation, indicating these equations as acceptable. Thus, the stepwise multiple regression analysis of the whole sample performed the final BIA prediction equation in both of the standing and supine BIA measure (Table 3).

### 3.3. The Final Standing and Supine BIA Prediction Equations for ASM

The multiple regression analysis of the whole sample performed the final BIA prediction equation in both of the standing and supine BIA measure (Table 3). The *RI*, the most essential predictive factor, explained *R*^2^ = 90.1% and *R*^2^ = 91.5% of variability in ASM and accumulated *R*^2^ (combined with *Xc* and sex in InBodyS10 and with Xc, Sex, and body weight in InBody770) explained variability up to 92.5% and 92.7% in the standing and supine BIA measure, respectively. The group predictive accuracy of SEE and subjective rating was following as 1.01~1.02 kg with the “excellent” rating in men and “good” or “very good” rating in women for the final standing and supine BIA prediction equation. The two newly developed equations for VIF showed no multicollinearity among variables. As shown in Figure 2A,B, there were no significant mean difference in ASM between DXA and BIA measurements in each BIA equation. The slope and intercept from the line of identity were not significantly different from 1 and 0 (*p* > 0.899). The Bland–Altman plots for the individual predictive accuracy are shown in Figure 3A,B. They showed no significant *R*_y-y’, mean_ = 0.138 (*p* > 0.05). The CE was − 0.02 ± 1.01 kg and 0.02 ± 1.00 kg with all no significant differences (*p* = 0.758 for the standing, *p* = 0.835 for the supine). The two standard deviations (±2 SD) of the limits of agreement were between 1.96 kg and −2.01 kg and between 1.98 kg and −1.95 kg, with the percentage of agreement individual (PAI) calculated into 80% and 84% for the standing and for the supine, respectively.

### 3.4. External Cross-Validation of Published and Built-in Equations for ASM

Table 4 compared the performance of ASM predicted by various BIA equations (including newly developed, built-in and published [13,14,15,16,17,18,19]) based on this study data with ASM measured by DXA. In the equation array of the standing and supine mode, there was a high determinant coefficient between each ASM_BIA_ and ASM_DXA_ (*R*^2^ = 0.891~0.923, *p* < 0.001) equations. However, the total error (TE) of all published and built-in equations exceeded the acceptable subjective range except for the built-in BIA_InBody770_ equation and BIA_Kyle_ equation. The BIA_Kyle_ equation shows a high *R*^2^ and its TE was within the acceptable subjective range of the standard error. In addition, it has no significant difference in the bias and PIA value was 74.4%. The built-in BIA_InBody770_ equation shows acceptable accuracy with a high *R*^2^, precise TE with subject ratings as “Good” for men and “Very good” for women, no difference in the bias, and 77.9% of PIA, whereas the built-in BIA_InbodyS10_ equation showed poor subject ratings with poor TE, significant difference in the bias and 37.4% of PIA. The two new BIA equations performed best in estimating the ASM with much higher *R*^2^ and TE with the acceptable subject ratings.

### 3.5. The Agreement of Sarcopenia between DXA-Measured and BIA-Predicted ASMI

The cut-offs for sarcopenia by AWGS classified the values of ASMI obtained from the four acceptable BIA equations and DXA measures into sarcopenia and normal as shown Table 5. The overall agreement of two new BIA prediction equations was statistically significant and rated as substantial, whereas that of the published BIA_Kyle_ and built-in BIA_InBody770_ was significantly rated as moderate. Those new BIA prediction equations had a specificity, a positive predictive value (PPV), and a negative predictive value (NPV) of 80% or more. The sensitivity was below 60% in all the acceptable BIA equations, except that the new BIA prediction equations had a sensitivity of 60.0% in the standing and 63.3% in the supine with its agreement as substantial.

## 4. Discussion

The ongoing need for BIA prediction equations that can be directly comparable to DXA in epidemiological and clinical settings [5,7,12] has led us to develop and cross-validate accurate BIA prediction equations in the standing and supine position for the ASM in elderly Korean men and women. The accuracy and precision of the newly established equations for ASM in this study were reasonable not only at the group level (*r*^2^ = 92.5~92.7%, SEE = 1.01 kg~1.02 kg) but also at the individual level (Bias = 0.01~0.02, 95% LOA = ±1.96~±1.98, PIA = 80~88%), thus indicating that these equations are valid and applicable to clinical settings as well as large-scale epidemiological studies. The sensitivity, specificity and overall agreement for the diagnosis of sarcopenia by the two new prediction equations were reasonably applicable to clinical diagnosis, except that a little caution on sensitivity is required.

The main achievement of the current study was to obtain two newly accurate prediction equations at the individual and population level. The new standing and supine BIA prediction equations included three-to-four dependent variables, consistent with the latest studies [18,21,26,27]. In particular, the independent variable *RI* explained 90.1% and 91.5% of the variability in our each equation, whereas in previous studies, the *RI* explained only 41.5% [19], 63.5% [19], 77.2% [14], 82.5% [17], 83.6% [21], 85.2% [14], 85.6% [18], and 88.3% [15] of the variability. Additional prediction variables further improved our prediction equations. Therefore, the group predictive accuracy, which was 92.5% and 92.7% of the variability and 1.01 kg and 1.02 kg of SEE, in the new standing and supine prediction equation, was highly more accurate than most previous published prediction equations (*R*^2^ = 75.7~90.0%, SEE = 1.22 kg~1.46 kg) [14,17,19,21]. It also was comparable to the result of *R*^2^ = 91.0% and 95.2% and SEE = 1.01 kg and 1.12 kg from Peniche and Kyle’s large-scale equation [16,18]. The main reasons for the improved group predictive accuracy (*R*^2^ and SEE) in this study may be due to the large sample size, sufficient magnitude of sample-to-predictor ratio, and using the high-frequency resistance (*R*) of 250 kHz. Generally, large samples (N = 100~400 subjects) are needed to ensure that the data are representative of the population for whom the equation was developed, and statisticians recommend a minimum of 20 to 40 subjects per predictor variable [23,24]. The new equations are based on a large sample of 199 in older Koreans and the ratio of the sample size to the number of predictors in this study was 50 to 67 subjects per predictor, which is larger than the recommended minimum ratio of 20 to 40 subjects per predictor. This large sample size and sufficient ratio may have led to have more stable regression weights for each predictor in the equation [23,24]. Meanwhile, the measurements of the resistance from 250 kHz are derived from the intracellular conductor as well as extracellular conductor in in the skeletal muscle. Unlike the model of the single-frequency (50 kHz) whole-body bioimpedance that predominately estimates the ASM from the extracellular conductor, predictive accuracy can be increased when estimating the ASM from the high-frequency 250 kHz bioelectrical impedance that conducts intracellular fluids of the skeletal muscle cells. Our findings showed the higher predictive accuracy of *R*^2^ and SEE from use of the high-frequency resistance (250 kHz) as opposed to the lower-frequency resistance 50 kHz (e.g., *R*^2^ = 91.5%, SEE = 1.09 kg ASM for the *RI* at 250 kHz vs. *R*^2^ = 90.6%, SEE = 1.49 kg for the *RI* at 50 kHz of InBodyS10). The result from this study is consistent with Segal’s findings that the measurements of the resistance from the high-frequency (100 kHz) can estimate both the intracellular fluid and extracellular fluids (i.e., total body water), with the highest accuracy among the 5k Hz and 100 kHz [26,27]. Therefore, the frequency-specific measurements that induce the suitable conductivity of the intracellular fluid in the skeletal muscle cells are considered to be very important factors to improve the accuracy of prediction equations. On the other hand, the individual predictive accuracy in this study showed no bias (CE), the allowable range of the limit of agreement, and the percentage of individual agreement (PIA) being more than 80%. Especially, compared to the LoA of ±2.12~±2.78 in the previous studies, the LoA of ±1.96~±2.04 in the present study was improved and suitable for predicting the group and individual level of ASM.

We performed external cross-validation of the published and built-in BIA prediction equations for ASM of Korean older population. The acceptable standard of predictive accuracy of those equations was *R*^2^ = 80% or more, no bias, no more than 1.45 kg SEE for men and no more than 1.16 kg SEE for women, and more than 70% for PIA. Our results indicated that the built-in BIA prediction equation of InBody770 and Kyle’s supine-position prediction equation was found to be acceptably accurate at the group and individual level. Conversely, previous published and built-in prediction equations only had good explanatory values (*R*^2^) and were found unacceptable for the SEEs and individual accuracy. The BIA_Kim_ and the BIA_Yamata_ regression equation developed for Korean or Asian older populations were found to be the least accurate in the present study, owing to low accuracy when they were initially developed. In fact, the TEs for two new BIA prediction equations in this study was 1.00~1.04 kg, and the TEs from BIA_Kim_ and BIA_Yamata_ were 5.91 kg and 1.86 kg, respectively. Therefore, both Kim and Yamata’s prediction equation for Korean or Asian populations should be selectively replaced by the new prediction equations put forth by the present study. Finally, the main reason for inaccuracies in other published prediction equations is the difference in body shape by population. Unlike the long and short torso of Caucasians and African-Americans, the short and long torso of Asians makes a different relation of muscle volume to current flow and resistance [11,12]. Therefore, the need to develop a population-specific prediction equation was confirmed again. Overall, the results of external cross-validation study showed that Kyle’s prediction equation and InBody770′s built-in prediction equation can perform accurate prediction at the group and individual levels for the Korean older population.

Lastly, we validated four equations, which were BIA_InBody770_new_, BIA_InBodyS10_new_, BIA_Kyle_ and BIA_InBodyS10_, to determine the applicability to the diagnosis of sarcopenia. Among the four equations, the two new standing and supine BIA prediction equations had the highest sensitivity, specificity, positive predictive value, negative predictive value, and overall agreement, which was higher than the size of the sensitivity and specificity of the BIA_Kyle_ and BIA_InBody770_ prediction equations. In this study, the overall agreement, sensitivity, specificity, PPV and NPV were also analyzed to determine whether ASMI by MF-BIA is feasible for individual testing and clinical application. All of the indices were very good, but the sensitivity was moderate, around 60%. The sensitivity of about 60% was higher than the previously reported 37~55% [27], but it still needs to be improved. One possible explanation for the low sensitivity in the present study is that MF-BIA-based ASMI in false-negative cases overestimated only 0.02 to 0.27 from the ASMI cutoff values of 7.00 in men and 0.06 to 0.49 from ASMI cutoff values of 5.40 in women (data not shown). When the newly developed MF-BIA prediction equations are applied in the clinical settings, ASMI that ranges from 7.02 to 7.27 in men and 5.46 to 5.89 in women should be considered as false-negative and perform an additional diagnosis by DXA. For other ASMI ranges, the results of ASMI by MF-BIA can be considered as accurate diagnostic results in clinical settings. However, further research is needed to improve sensitivity. In future studies, it would be effective to increase the explanatory power of the skeletal muscle impedance by passing the current of the intracellular fluid using a higher frequency [11,12,13,23,28,29]. Therefore, sensitivity should be taken this into account in clinical settings.

Two new BIA equations were developed based on 199 healthy elderly people aged above 70 and living in Korean communities. Given that the sample size of the population under study, the specific age group, the living environment and health status of subjects could not represent the entire population, this may affect the generalizability of our results. In this research, DXA was used as the reference standard for the assessment of ASM. It should be noted that DXA has some limitations with respect to ASM measurement technology, in which ASM was measured as the sum of muscle mass calculated by subtracting the mineral content from the lean weight of arms and legs (limbs). DXA did not separate skeletal muscle from the skin, connective tissue and blood vessels [30,31]. DXA may overestimate the ASM [31,32] of the human body compared with the golden standard (for example, CT and MRI) for quantifying body composition. Although it has been proven that there was a high correlation between DXA and golden standard, and DXA has been permitted as the reference standard for the BIA method [5,7,10,33,34] with the penetration of fat into muscle tissue and the partial covering of age related muscle mass loss through the replacement of extracellular water and connective tissue [28,30], these factors may affect the accuracy of prediction results.

## 5. Conclusions

The newly developed standing and supine BIA prediction equations are quite acceptable for individual and group prediction accuracy, sensitivity, specificity, and overall agreement, making them useful in epidemiological and field settings as well as for the clinical diagnosis of sarcopenia. Future research is needed to improve the sensitivity of diagnosis of sarcopenia.

## Figures and Tables

**Figure 1 ijerph-17-05847-f001:**
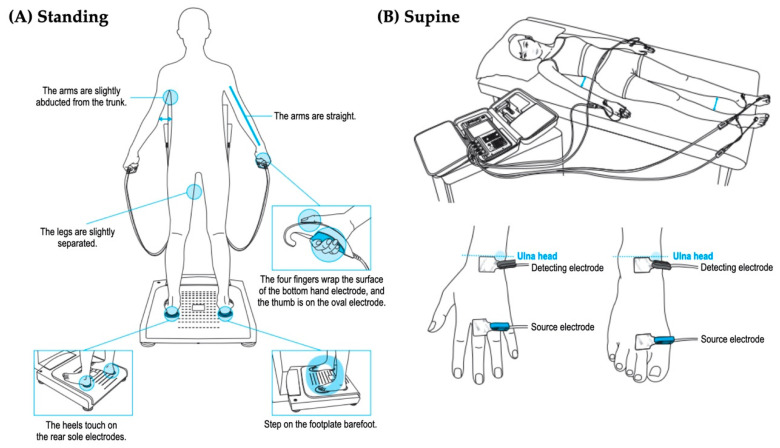
The testing postures and the electrode placements (**A**) BIA in the standing position (**B**) BIA in the supine position [permitted from the manufacture].

**Figure 2 ijerph-17-05847-f002:**
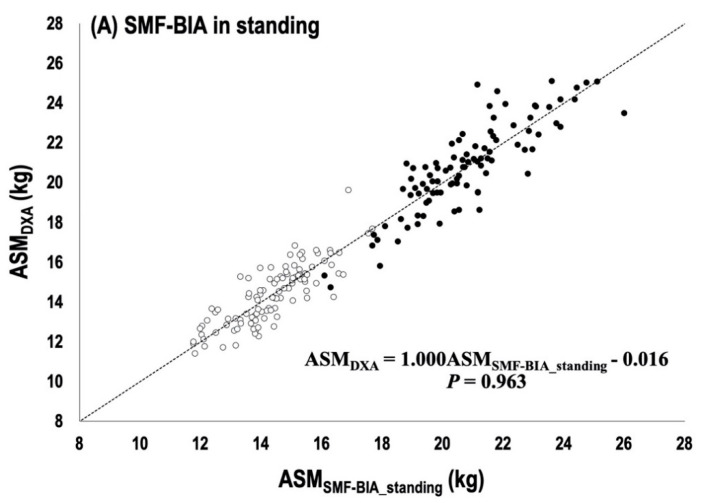
Bivariate regression analysis; (**A**) The line of best fit and standard error of estimate for the standing BIA prediction equation (**B**) The line of best fit and standard error of estimate for the supine BIA prediction.

**Figure 3 ijerph-17-05847-f003:**
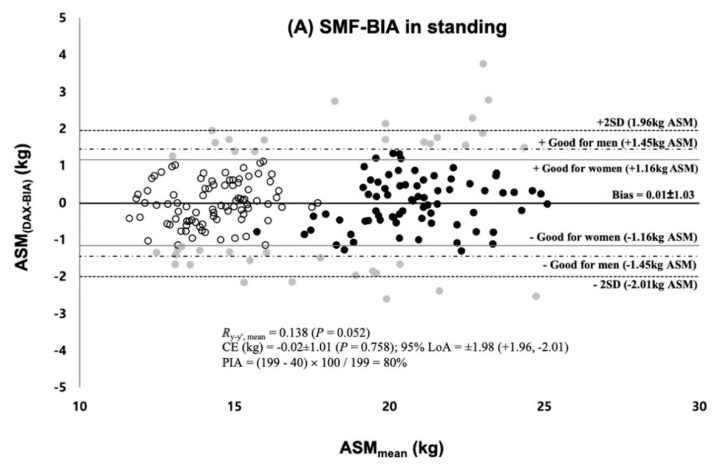
Bland–Altman plot of residual scores and mean difference between the measured and predicted ASM; (**A**) Residuals and mean difference from the standing BIA prediction equation (**B**) Residuals and mean difference from the supine BIA prediction equation; *R*_y-y’, mean_ = concordance correlation coefficient between the residuals and the means of the measured ASM and predicted ASM; LoA = Limits of Agreement in ±1.96 SD; PIA = Percentage of individual agreement; 
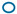
 = women within 1.16 kg ASM, 
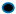
 = men within 1.45 kg ASM, 
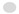
 = individuals who are out of the minimum acceptable standard for prediction errors (i.e., good rating) [11,24,25].

**Table 1 ijerph-17-05847-t001:** General characteristics of the participants for development and validation of equations.

Variables	Development Group		Cross-Validation Group	
	Men (*n* = 63)	Women (*n* = 70)	Men (*n* = 31)	Women (*n* = 35)
Age (years)	76.4 ± 4.2	76.1 ± 4.1	75.9 ± 4.1	75.6 ± 4.3
Height (cm)	166.5 ± 5.1	152.7 ± 5.0 *	167.6 ± 4.3	153.3 ± 4.2 *
Weight (kg)	65.6 ± 7.7	55.4 ± 5.8 *	66.0 ± 6.8	54.8 ± 7.3 *
BMI (kg·m^−2^)	23.7 ± 2.3	23.8 ± 2.2	23.5 ± 2.3	23.3 ± 2.7
FFM (kg)	50.2 ± 4.6	36.9 ± 3.1 *	51.1 ± 4.0	37.0 ± 3.9 *
FM (kg)	15.7 ± 5.3	17.3 ± 4.6 *	15.1 ± 5.1	18.6 ± 4.1 *
PBF (%)	23.2 ± 6.9	32.5 ± 4.9 *	22.4 ± 5.0	31.5 ± 5.5 *
ASM (kg)	20.6 ± 2.4	14.4 ± 1.4 *	21.1 ± 2.0	14.3 ± 1.8 *
*Standing mode of BIA*				
*R* _@50kHz_	528 ± 55	616 ± 50 *	532 ± 42	663 ± 61 *
*R* _@250kHz_	480 ± 50	564 ± 46 *	484 ± 39	571 ± 57 *
*Xc* _@5kHz_	24.5 ± 4.6	24.8 ± 4.0	24.7 ± 3.6	24.9 ± 4.1
*Xc* _@50kHz_	47.2 ± 7.4	48.1 ± 6.1	47.5 ± 4.7	49.2 ± 6.5
*RI* _@50kHz_	53.1 ± 6.7	38.1 ± 3.9 *	53.1 ± 4.8	38.1 ± 4.5 *
*RI* _@250tand_	58.4 ± 7.3	41.6 ± 4.3 *	58.4 ± 5.3	41.6 ± 5.0 *
*Supine mode of BIA*				
*R* _@50kHz_	488 ± 49	575 ± 46 *	487 ± 41	582 ± 57 *
*R* _@250kHz_	438 ± 44	521 ± 41 *	437 ± 38	527 ± 54 *
*Xc* _@5kHz_	24.3 ± 4.7	25.1 ± 4.3	25.0 ± 2.9	25.0 ± 3.6
*Xc* _@50kHz_	47.9 ± 7.2	49.3 ± 6.9	47.9 ± 4.5	50.0 ± 5.7
*RI* _@50kHz_	57.3 ± 40.8	40.8 ± 4.0 *	58.1 ± 5.3	40.8 ± 4.8 *
*RI* _@250tand_	64.0 ± 7.9	45.0 ± 4.4 *	64.8 ± 6.0	45.1 ± 5.5 *

BMI: body mass index; FFM: Fat-free mass; FM: Body fat mass; PBF: percent body fat, ASM: appendicular skeletal muscle mass; *R*, *Xc*, and *RI*: resistance, reactance, and resistance index at 5 kHz, 50 kHz and 250 kHz, respectively, * = significantly different from men at *p* < 0.05, ^‡^ = significantly different from the development group at *p* < 0.05.

**Table 2 ijerph-17-05847-t002:** Development and validation of predictive bioimpedance analysis (BIA) equations for appendicular skeletal muscle mass (ASM) on Korean older people.

**Standing Mode of SMF-BIA**		
	*Development group (n = 133)*	*Cross-validation group (n = 66)*
Measured ASM	17.3 ± 3.66 kg	17.5 ± 3.92 kg
ASM prediction equation	0.273RI@250 kHz + 1.369sex + 0.049Xc@50 kHz + 0.032 BW − 1.118	
	^‡^*R*^2^ = 0.923, SEE = 1.10 kg, CV = 6.4%,	
	SR(M) = Very good, SR(W) = Good	
	VIF: *RI* = 6.88, *Xc* = 1.45, BW = 2.74,	
	sex = 3.87	
Predicted ASM	17.5 ± 3.73 kg	17.6 ± 3.73 kg, * *p* = 0.693
		*R*^2^ = 0.934, TE = 1.00 kg, CV = 5.7%
		SR = Excellent (M), SR = Very good (W)
**Supine Mode of SMF-BIA**		
	*Development group (n = 133)*	*Validation group (n = 66)*
Measured ASM	17.3 ± 3.66 kg	17.5 ± 3.92 kg
ASM prediction equation	0.266RI@250 kHz + 1.227sex + 0.057*Xc*_@5kHz_ + 0.960	
	^‡^*R*^2^ = 0.919, SEE = 1.06 kg, CV = 6.1%,	
	SR(M) = Very good, SR(W) = Good	
	VIF: *RI* = 4.34, *Xc* = 1.40, sex = 3.73	
Predicted ASM	17.3 ± 3.51 kg	17.4 ± 3.58 kg, * *p* = 0.291 *R*^2^ = 0.948, TE = 0.93 kg, CV = 5.3%
		SR = Excellent (M), SR = Very good (W)

*RI* = resistance index; *Xc* = reactance; sex: (M)en = 1, (W)omen = 0; BW = body weight; *R*^2^ = determinant of coefficient; ^‡^
*R*^2^ = Adjusted *R*^2^; CV = coefficient variation; VIF = variation inflation factor; SEE = standard error of the estimate; SR = subject rating of standard for prediction error [ideal = 0.72~0.90(M), 0.54~0.65(W); excellent = 0.90~1.09(M), 0.65~0.83(W); very good = 1.09~1.27(M), 0.83~1.01(W); good = 1.27~1.45(M), 1.01~1.16(W); fairly good = 1.45~1.63(M), 1.16~1.30(W); fair good = 1.45~1.63(M), 1.16~1.30(W); Fair = 1.63~1.81(M), 1.30~1.44(W); poor = >1.81(M), >1.44(W); unit = kg ASM] [11,24,25]; TE = Total Error; * = *p*-value of paired *t*-test for the mean difference between measured and predicted means.

**Table 3 ijerph-17-05847-t003:** The final predictive standing and supine BIA equations for ASM on Korean older people.

Final Prediction Equations	
*Standing Mode of SMF-BIA (n = 199)*	
Measured ASM	17.4 ± 3.74 kg
ASM prediction equation	0.286RI@250 kHz + 1.367sex + 0.054Xc@50 kHz + 0.031 BW − 1.864
	^‡^*R*^2^ = 0.925, SEE = 1.02 kg, CV = 5.9%, SR = Excellent (M), SR = Good (W)
	VIF: *RI*_@250kHz_ = 7.48, sex = 4.04, *Xc*_@50kHz_ = 1.41, BW = 2.91
Predicted ASM	17.4 ± 3.60 kg, * *p* = 0.758
*Supine Mode of SMF-BIA (n = 199)*	
Measured ASM	17.4 ± 3.74 kg
ASM prediction equation	0.276RI_@250kHz_ + 1.151sex + 0.059Xc@5 kHz + 0.429
	^‡^*R*^2^ = 0.927, SEE = 1.01 kg, CV = 5.8%, SR = Excellent (M), SR = Very good (W)
	VIF: *RI* = 3.91, *Xc*= 1.11, sex = 3.73
Predicted ASM	17.4 ± 3.60 kg, * *p* = 0.835

*RI* = resistance index; *Xc* = reactance; sex: (M)en = 1, (W)omen = 0; BW = body weight; ^‡^
*R*^2^ = adjusted determinant of coefficient; CV = coefficient variation; VIF = variation inflation factor; SEE = standard error of the estimate; SR = subject rating of standard for prediction error [ideal = 0.72~0.90(M), 0.54~0.65(W); excellent = 0.90~1.09(M), 0.65~0.83(W); very good = 1.09~1.27(M), 0.83~1.01(W); good = 1.27~1.45(M), 1.01~1.16(W); fairly good = 1.45~1.63(M), 1.16~1.30(W); fair good = 1.45~1.63(M), 1.16~1.30(W); Fair = 1.63~1.81(M), 1.30~1.44(W); poor = >1.81(M), >1.44(W); unit = kg ASM] [11,24,25]; * = *p*-value of paired *t*-test for the mean difference between measured and predicted means.

**Table 4 ijerph-17-05847-t004:** External cross-validation of BIA equations and devices for ASM measured by dual-energy X-ray absorptiometry (DXA).

Device	ASM (Mean ± SD)	*R* ^2^	TE (kg)	Subjective Rating		CE (Mean ± SD)	LoA (Kg)	*r* _y-y’,mean_	PIA
				Women	Man				
DXA	17.38 ± 3.74								
*Standing Modes of BIA*									
BIA_standing_New_	17.39 ± 3.59	0.924	1.04	Good	Excellent	− 0.02 ± 1.03	−2.04, 2.01	−0.145 *	81.4
BIA_InBody770_	17.35 ± 4.00	0.917	1.15	Good	Very good	0.03 ± 1.15	−2.22, 2.29	−0.223 *	77.9
BIA_Yamada_	18.67 ± 4.07	0.891	1.86	Poor	Poor	−1.29 ± 1.35 **	−3.94, −1.35	−0.252 **	48.7
*Supine Modes of BIA*									
BIA_supine_New_	17.37 ± 3.60	0.928	1.00	Very good	Excellent	0.02 ± 1.10	−1.95, 1.98	0.138	83.9
BIA_InBodyS10_	19.08 ± 4.43	0.914	2.20	Poor	Poor	−1.71 ± 1.38 **	−4.42, 1.00	−0.464 **	37.4
BIA_Vermeiren_	15.80 ± 3.38	0.916	1.81	Poor	Poor	1.42 ± 1.13 **	−0.80, 3.64	−0.327 **	39.7
BIA_Scaroflieri_	17.77 ± 3.46	0.906	1.21	Fairly good	Very good	−0.39 ± 1.16 **	−2.66, 1.89	−0.243 **	74.9
BIA_Sergi_	16.60 ± 3.45	0.919	1.33	Fair	Good	0.78 ± 1.08 **	−1.34, 2.90	−0.275 **	67.3
BIA_Kyle_	17.34 ± 4.09	0.923	1.15	Good	Very good	0.04 ± 1.16	−2.23, 2.30	−0.307 **	74.4
BIA_Kim_	11.64 ± 2.79	0.899	5.91	Poor	Poor	5.75 ± 1.42 **	2.96, 8.53	−0.098	0.0
BIA_Rangel_	16.81 ± 4.06	0.919	1.30	Fairly good	Good	0.57 ± 1.17 **	−1.72, 2.87	−0.276 **	64.3

ASM = appendicular skeletal muscle mass; *R*^2^ = determinant of coefficient between ASM_DXA_ and ASM_BIA_; TE = Total; Limits of agreement were calculated as mean difference ± 1.96 times SD; *r*_y-y’,mean_ = concordance Pearson correlation coefficient between differences (ASM_DXA_ − ASM_BIA_) and means ((ASM_DXA_ + ASM_BIA_)/2); SR = subject rating of standard for prediction error [ideal = 0.72~0.90(M), 0.54~0.65(W); excellent = 0.90~1.09(M), 0.65~0.83(W); very good = 1.09~1.27(M), 0.83~1.01(W); good = 1.27~1.45(M), 1.01~1.16(W); fairly good = 1.45~1.63(M), 1.16~1.30(W); fair good = 1.45~1.63(M), 1.16~1.30(W); Fair = 1.63~1.81(M), 1.30~1.44(W); poor > 1.81(M), > 1.44(W); unit = kg ASM] [11,24,25]; PIA = Percentage of individual agreement, * *p* < 0.05; ** *p* < 0.001.

**Table 5 ijerph-17-05847-t005:** Prevalence, sensitivity and specificity of the acceptable BIA equations to determine sarcopenia.

Equations/Device	Overall Agreement *N* (%)	Cohen’s Kappa	Sensitivity	Specificity	PPV	NPV
*Standing Modes of BIA*						
BIA_InBody770_NEW_	184 (92.5)	0.664 *	60.0	98.2	85.7	93.3
BIA_InBody770_	165 (82.9)	0.397 *	51.5	89.2	48.6	90.2
*Supine Modes of BIA*						
BIA_InBodyS10_NEW_	185 (93.0)	0.691 *	63.3	98.2	86.4	93.8
BIA_Kyle_	168 (84.4)	0.416 *	48.5	91.6	53.3	89.9

PPV = positive prediction value; NPV = negative prediction value; AWGS Cut-off of ASMI: Female < 5.4 kg·m^−2^, Male < 7.0 kg·m^−2^, Agreement is poor if *k* < 0.00, slight if 0.00 < *k* < 0.20, fair if 0.21 < *k* < 0.40, moderate if 0.41 < *k* < 0.60, substantial if 0.61 < *k* < 0.80, and almost perfect if *k* > 0.80; * *p* < 0.001 [25].

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
