# Peer review of "Prediction Equations of the Multifrequency Standing and Supine Bioimpedance for Appendicular Skeletal Muscle Mass in Korean Older People"

_ijerph, 2020, doi:10.3390/ijerph17165847_

Round 1
Reviewer 1 Report
In this study, authors have proposed two novel prediction equations for the measurement of appendicular skeletal muscle through the use of bioimpedance analysis in the older Korean population. Moreover, the proposed prediction equations were compared with published prediction equations based on literature findings. 199 healthy older adults were enrolled in the experimental protocol.
This manuscript is well written and conveys a clear message. However, some major revision should be considered to improve the significance of the paper.
The measurement of appendicular skeletal muscle through the use of bioimpedance analysis is not exactly a novelty over the past years. As the authors reported several research efforts have been spent in the development of novel prediction equations for the analysis of sarcopenia in older people. In the introduction section, authors should clarify the novelty of the proposed method in comparison whit previous literature studies, more specifically from those who involve the Korean population such as:
Kim, Jung Hee, et al. "Assessment of appendicular skeletal muscle mass by bioimpedance in older community-dwelling Korean adults." Archives of gerontology and geriatrics 58.3 (2014): 303-307.
In addition, in the introduction section, authors should better explain the need of assessing different prediction equations for standing and supine conditions. How authors envision the applicability of those different scenarios in the clinical setting?
In the material and methods section, an image on the electrode placement on the anatomical side should be added. Moreover, reporting acronyms, such as SEE, CV, or VIF, without explaining the meaning could be misleading (see lines 159-161). When acronyms are first mentioned, the extended form and the relative formula should be added.
In lines, 133-136, please add references to each study.
In the results section, line 218: “ The Bland-Altman plots .. are shown in Figure 1(B) & (C)”. Please correct this sentence with: “ The Bland-Altman plots .. are shown in Figure 1(B) & (D)”. Please improve the description and visibility of Figure 1. It is not properly explained the meaning of black and white dots and the sizing of the figure does not allow a clear text understanding.
Reviewer 2 Report
Authors validated the bioimpedance analysis for the prediction of ASM in 199 Korean older adults. Methodology was properly designed and the paper is generally well structured. My only main concern is related to the novelty of the study; in fact it seems that authors just used already known methods and instrumentations and applied them to a different cohort of subjects (i.e. Korean). Authors should stress the novelties and emphasize the scientific impact of the paper. Otherwise, it is more suitable for a conference rather than a journal paper due to the scientific impact.
Other issues:
- Please always report the full name at the first appereance of the acronyms (for example, ASM line 60, AWGS line 82)
- MRI does not involve exposition to radiation, please clarify in line 54.
- Please use the reccomandations reported in the GUM for the writing of number and unit of measure: a space between the two entities must be added.
- Please check the english of lines 104-105 and rewrite the sentence. Similar for lines 126-128
- I strongly suggest to add a figure in which the electrode placement can be observed.
- Line 198, there is a red dot, please modify.
- All the indices reported in the table caption must be introduced in the methods with the relative meaning and equation.
- Authors achieved a sensitivity value of approximatively 60. This is not a good result for the real application of the method. Authors should deeply discuss this outcome, by highlitghting possible reasons and corrections and not limited the comments to "little caution is required".
Reviewer 3 Report
This study sets to develop and validate equations to predict appendicular skeletal muscle mass in the standing and supine positions. 199 healthy older adults completed the measurements of multifrequency BIA and dual-energy X-ray absorptiometry (DXA). The BIA prediction equations appear to be acceptable. The statistical methodology is sound. The results are reasonable. The residual plots in figure 1 are very helpful and convincing. Overall the paper is well written.
Round 2
Reviewer 1 Report
All requirements have been met.
Reviewer 2 Report
Authors have properly answered to my previous comments and it is now clear the novelty of the manuscript.